# Mediterranean Food Industry By-Products as a Novel Source of Phytochemicals with a Promising Role in Cancer Prevention

**DOI:** 10.3390/molecules27248655

**Published:** 2022-12-07

**Authors:** Andrea Agaj, Željka Peršurić, Sandra Kraljević Pavelić

**Affiliations:** 1Faculty of Medicine, Juraj Dobrila University of Pula, Zagrebačka 30, 52100 Pula, Croatia; 2Faculty of Chemical Engineering and Technology, University of Zagreb, Trg Marka Marulića 19, 10000 Zagreb, Croatia; 3Faculty of Health Studies, University of Rijeka, Ul. Viktora cara Emina 5, 51000 Rijeka, Croatia

**Keywords:** by-products, anticancer activity, polyphenols, cancer, Mediterranean food industry

## Abstract

The Mediterranean diet is recognized as a sustainable dietary approach with beneficial health effects. This is highly relevant, although the production of typical Mediterranean food, i.e., olive oil or wine, processed tomatoes and pomegranate products, generates significant amounts of waste. Ideally, this waste should be disposed in an appropriate, eco-friendly way. A number of scientific papers were published recently showing that these by-products can be exploited as a valuable source of biologically active components with health benefits, including anticancer effects. In this review, accordingly, we elaborate on such phytochemicals recovered from the food waste generated during the processing of vegetables and fruits, typical of the Mediterranean diet, with a focus on substances with anticancer activity. The molecular mechanisms of these phytochemicals, which might be included in supporting treatment and prevention of various types of cancer, are presented. The use of bioactive components from food waste may improve the economic feasibility and sustainability of the food processing industry in the Mediterranean region and can provide a new strategy to approach prevention of cancer.

## 1. Introduction

Food industry waste is a global problem. In 2017, approximately 37 million tons of agricultural by-product waste was generated worldwide [1]. These by-products consist of peel, stem, seeds, and pulp, among others. Improper disposal of such by-products might negatively impact the environment, but on the other hand, these by-products have the potential for reuse in health applications, and they are cheap and easily accessible [2,3,4]. For example, phenolic compounds with beneficial properties for human health are present in such food waste [5]. Plant-derived extracts could also be exploited in an integrative approach to cancer prevention and integrative cancer treatment. Non-edible parts such as peels or seeds, indeed, have a similar content of polyphenols as the edible parts, providing anticancer properties and possibly being easily extracted [6]. Therefore, valorisation of agro-industrial residues as a source of phenolic compounds or other bioactive compounds is at hand [7,8,9]. Technological developments in this area are, however, necessary as a loss of compounds caused by inadequate processing with conventional extraction methods (maceration, Soxhlet, shaking, etc.) may lead to the destruction of valuable bioactive molecules [10,11].

Currently, the Mediterranean diet, even though difficult to be exactly defined, may be regarded as a best evidence-based, healthy and sustainable diet [12]. The composition of the Mediterranean diet, always going along with a specific lifestyle, is particularly based on olive oil, grains, legumes, fruits and vegetables consumption. This diet includes only moderate quantities of processed foods as well, contrary to many other dietary styles, which arebased on heavily processed food. In addition, some country-specific variations in the overall dietary pattern of the Mediterranean diet depend on local cultures and traditions as well [13]. The Mediterranean diet is characterized by positive, health-promoting, and nutritional benefits, as extensively evidenced in the literature dedicated to this topic since 1960. Even though the Mediterranean diet can be classified as sustainable, the production of well-known Mediterranean food products generates substantial amounts of waste. This is particularly true for wine production. It is estimated that 25 kg of waste is produced for every 100 kg of grapes [14]. This raises an economic and environmental challenge for the wine industry [15,16]. Industrial winemaking activities produce solid residues such as grape pomace (60% of total wine by-products), mainly containing grape skins (50%), pulp, stalks (25%), and seeds (25%) [17]. Furthermore, wine lees are produced during fermentation and account for 25% of the waste produced during winemaking as well [18,19]. All of these by-products are a rich source of dietary fibres and antioxidant compounds [20,21]. Consequently, wine by-products can be a valuable material, as compounds found in these by-products have already been associated with wide health-promoting properties, including the improvement of digestive tract disorders and anticancer activity [22]. These wine by-products ingredients may be used for production of food with functional properties or for development of nutraceuticals. This is just one example of the food industry that generates by-products in the Mediterranean region. Many by-products are generated worldwide and that have a high potential for re-usage due to possible health benefits. In this review, different by-products and their possible usages are discussed in detail with an emphasis on use of by-products from the Mediterranean food industry as a source of phytochemicals with anticancer activity. The by-products presented herein from the Mediterranean food industry are selected from typical Mediterranean food production lines and their wide representation in the Mediterranean region. 

## 2. Valorisation of Food Waste as a Source of Antioxidant Phenolic Compounds

Phenolic compounds are secondary metabolites generated by all plants. Even though these compounds have been consumed since ancient times, due to their widespread presence in food, the knowledge on their beneficial medicinal properties is relatively new. For instance, the first known use of gallic acid dates back to 1788, when it was used for photographic purposes, whereas today, it is widely used as a standard for quantifying total phenolics of numerous extracts [23]. In recent decades, fascinating biological properties have been documented for this group of compounds [24,25]. In particular, their protective function due to antioxidant and antimicrobial properties has been studied. In addition, other beneficial effects on human health such as antitumor, antiproliferative, anti-inflammatory, anti-obesity, and even ultraviolet (UV) radiation protective activity has been documented as well [26]. These properties have already been acknowledged in veterinary applications and animal health, since some phenolic compounds, such as tannins, can be used as an alternative to oral antibiotics given to livestock [27]. On the other hand, the cytotoxicity of these compounds to normal human cells is very low, which makes them very attractive for use in humans [28].

The extraction of bioactive compounds from food waste can be performed by conventional and non-conventional extraction procedures, where the latter have been recently reviewed and tested extensively as they are more environmentally friendly and often give a better yield or even improved quality of the extracts in comparison with conventional methods. Extraction processes are generally based on method of choice adjusted to the raw material used for extraction and the organic solvent. It is now well established that the conventional techniques require large amounts of organic solvents and are time consuming, which correlates with energy consumption as well. The non-conventional techniques are being growingly introduced, accordingly, as cleaner and less energy-consumption methods. They may reduce or eliminate the use of toxic organic solvents which helps in the preservation of the natural environment [29,30]. Many of the current technologies applied in food waste reusage of phenolic compounds have certain limitations. For instance, the use of organic, often toxic, solvents for extraction of bioactives has a negative impact both on human health and the environment if it is not used in a safe manner or properly disposed [31]. Furthermore, most of the employed extraction technologies have low efficiency, require long extraction times and high energy consumption, as well as cause hydrolytic compounds degradation [32]. Because of these problems, it is crucial to find alternative techniques with reduced environmental impact and increased technological performance. 

Important non-conventional methods that may substantially decrease the footprint of the extraction workflow, including, for example, microwave-assisted extraction, ultrasound assisted extraction, enzyme digestion, and supercritical fluids [29]. Accordingly, the “green extraction” alternatives have been increasingly studied [33]. The application of green recovery technologies avoids harmful effects of conventional technologies and obtained extracts can be recycled into the food chain as functional additives for different products and applications. A number of researchers have suggested alternative extraction methods, such as microwaves, ultrasonic, electrical technologies (pulsed electric fields, high electric discharges, and ohmic heating) and mechanical treatments (pressurized hot water extraction and subcritical fluid extraction) [34]. These processes are suitable for improved recovery of valuable compounds since they use green solvents and reduce solvent consumption. Green solvents are environmentally friendly solvents, which are derived from the processing of agricultural crops [35]. One of such extraction methods for polyphenols is the Ultrasound Assisted Extraction (UAE), that has been successfully used for extraction thermolabile compounds. UAE causes disruption of cells’ membrane wall by compression and release of rarefaction waves, which facilitates the release of extractable biocompounds and enhances the heat and mass transfer. UAE can be considered as an eco-friendly technology with many other advantages as well [36].

### Use of Phenolic Compounds as Functional Food Ingredients

Synthetic additives in food are known to have negative impact on health; i.e., they may cause allergies or even bear carcinogenic effects [37]. Accordingly, alternatives are continuously sought for food storage improvements. Phenolic compounds are, for example, suitable candidates for the substitution of synthetic additives. These natural compounds can be used as preservatives, thanks to their antioxidant and antimicrobial activity, or can be used as flavouring agents to improve the organoleptic characteristics of products, or can be exploited as natural colourants, such as natural pigments anthocyanins widely spread in the plant world [38].

In addition, many manufacturers in global industries are focused on producing functional food enriched with phenolic compounds. Still, their mechanism of action, more specifically the pathways through which phenolic compounds act in the body, remain uncovered in detail [39]. Another information gap in the administration of phenolic compounds is the determination of the dose–response relationship as well as their potential systemic toxicology. These pharmacological data are needed to protect consumers and producers who use large amounts of phenolic compounds in their products [40]. Although further research is needed, several studies show promising results on their use in the therapeutic field [41].

## 3. Cancer and the Mediterranean Diet Protective Role

Cancer is a heterogeneous multi-stage disease, driven by progressive genetic and epigenetic changes. Epigenetic changes, as part of malignant transformation of cells, are known to be mostly influenced by lifestyle and nutrition [42]. Diet can, accordingly, have an important role in development and progression of cancer. One of the most promising diets with anticancer properties is the Mediterranean diet [43]. Numerous studies have highlighted the positive correlation between the Mediterranean diet and longevity, so it can be expected that people who adhere to such a dietary style have a longer life expectancy [44]. The Mediterranean diet has already shown an inverse association with metabolic disease, cardiovascular pathologies, and cancers [43]. Given its protective effects in the reduction in oxidative and inflammatory processes in cells and prevention of DNA damage, cell proliferation, and survival, angiogenesis and metastasis, the Mediterranean diet is considered an effective and manageable method to reduce cancer [45].

The Mediterranean diet is characterized by: a high consumption of whole grains (about 50–60% of the total caloric intake), and a high consumption of vegetables, fruit and legumes; the use of extra virgin olive oil that covers about 70% of the lipid supply; regular consumption of fresh fish (especially oily fish); regular but moderate consumption of red wine during main meals; an optimal supply of omega-3 both of animal and vegetable origin [46,47]. In particular, the added value comes from the diversity of healthy foods and their numerous nutritional benefits, as well as from the attention to the seasonality of each ingredient and the freshness of products to maximize the content of protective substances ([48,49,50]. The protective effect of the Mediterranean diet against many diseases, including cancer, is due to the high levels of antioxidants and anti-inflammatory compounds in the food [51]. Specifically, the Mediterranean diet has a high concentration of polyphenols available in olive oil, wine, and vegetables. For example, the consumption of extra virgin olive oil is correlated with low incidence of arteriosclerosis, diabetes, inflammatory and autoimmune diseases, skin diseases, ageing, and tumour pathology [52]. Polyphenols isolated from extra virgin olive oil inhibit the expression of the HER-2/neu protein (ERBB2), which is correlated with aggressive breast carcinoma [53]. Additionally, fruits and vegetables from the Mediterranean diet are rich in carotenoids, vitamins, and flavonoids, phytochemicals known for their antioxidant properties that prevent DNA damage [54]. Finally, omega-3 from fish and nuts may slow down cancer development and cancer cell proliferation, survival, angiogenesis, inflammation, and metastasis [55]. In addition, the microbial characteristics of individuals consuming a Mediterranean diet are thought to be protective against cancer [46]. Interestingly, the abundance of the *Fusobacterium nucleatum* in the intestinal microbiome that is associated with development of human colorectal carcinomas, was shown to increase after only two weeks of a high-fat and low-fibre Western-style diet intervention [56]. On the other hand, the microbiota of the people consuming the Mediterranean diet, is primarily characterized by a larger biodiversity, which positively impacts human health [57]. Higher adherence to the Mediterranean diet, monitored with the MedDietScore, was also correlated with decreased presence of *E. coli*, increased total presence of bacteria, higher *Bifidobacteria*/*E. coli* ratio and increased presence of short chain fatty acids (SCFAs) [58]. SCFAs have been shown to inhibit cancer cell growth to varying extents and through multiple mechanisms. Additionally, a meta-analysis conducted by [59] on diet and intestinal disease related to inflammation and cancer highlights the unique characteristics of bacterial population associated with the Mediterranean diet. The microbiota of subjects that consumed Mediterranean diet was enriched with beneficial bacteria that promote an anti-inflammatory environment. These results suggest that the effect of the Mediterranean diet on the gut microbiota has the potential to prevent cancer and other inflammation-related diseases of the gut. The complex protective role of Mediterranean diet against non-communicable diseases is briefly presented in Figure 1.

Epidemiological and clinical studies provide evidence on association between certain nutrition patterns and development or progression of different cancer malignancies such as colon, breast, and prostate cancer which describes these tumours as dietary-associated cancers as well [60,61]. A diet type plays a crucial role in colorectal cancer, as unhealthy eating habits are considered among the most important risk factors for a malignant transformation of cells. Generally, a better quality of diet alone is associated with lower colorectal cancer risk [62], independently of hereditary, traditional or environmental risk factors [63]. Adherence to the Mediterranean diet for example, has been shown to reduce colorectal cancer risk by approximately 30% and 45% in men and women, respectively. The percentage of risk also decreased significantly which has been assessed by examination of tumour sites, distal colon, and rectum [64]. The high protective effect of the Mediterranean diet was also confirmed in a recent study, which showed that adherence to a Mediterranean diet can reduce mortality rates in colorectal cancer patients. When the variables of gender and age are taken into account, the mortality rate could be reduced by 11% and 12% in men and women, respectively [65]. The Mediterranean diet is considered to be a protective factor in the decrease in breast cancer incidence, thanks to a regular intake of fibre, flavonoids, vitamins and carotenoids [66]. Several studies have investigated the association between the Mediterranean diet and the incidence of breast cancer. The results have shown that in post-menopausal women, the incidence of breast cancer decreases significantly (about 40%) when they follow a Mediterranean lifestyle [45]. Furthermore, it is demonstrated that a prostate cancer incidence was less widespread in Mediterranean area than in the Northern Europe [67,68]. High Mediterranean diet adherence was not only inversely associated with a low incidence of prostate cancer, but it was also associated with lower cancer malignancy [69] and mortality in patients without metastasis [67]. Overall, a high Mediterranean diet score was associated with up to a 78% reduced risk of prostate cancer in subjects with the highest Mediterranean diet score [70,71]. The strong protective effects elicited by the Mediterranean diet on many cancers are due to the presence of bioactive compounds contained in Mediterranean diet typical foods. However, these bioactive dietary components with antineoplastic properties are also significantly represented in by-products obtained during the production of these foods. The potential of by-products as a sustainable alternative source of compounds with anticancer potential will be elaborated in the following section.

## 4. Functional Properties of Mediterranean By-Products with Emphasis on Anticancer Activity

### 4.1. Wine Production—By-Products

Food industry waste is a global problem, and approximately 37 million tons of agricultural by-products were generated worldwide in 2017. By-products from industrial winemaking are wine lees and grape pomace composed of grape skins, pulp, stalks and seeds. In addition, by-products leaves and wine shoot can be formed during the winemaking process [72]. Although these by-products can cause important environmental and economic issues, the high content of bioactive compounds and dietary fibres makes these by-products an attractive source for further exploitation. Analysis of polyphenols by high pressure liquid chromatography (HPLC) with diode array detector (DAD) and coupled to mass spectrometry (MS) showed that grape skin is a rich source of anthocyanins, hydroxycinnamic acids, flavanols, and flavonol glycosides, whereas flavanols were mainly present in the seeds, large variations of the particular compound content were observed depending on the cultivar and vintage [73]. Further on, the prebiotic effect of dietary fibres contained in these waste by-products is probably the most important functional property for further exploitation. Dietary fibre reaches the colon, where it is fermented by the intestinal microbiota, generating SCFA, such as butyric, propionic, and acetic acids [74]. These compounds are associated with a wide range of physiological properties, including the improvement of digestive tract disorders and anticancer effects [75,76]. Food industry by-products, especially in the Mediterranean area, can be considered as sources of high-quality dietary fibres for food applications with good functional characteristics. However, further studies on the structures of these by-products and its integration in wide variety of foods needs to be explored [1].

#### Anticancer and Antimutagenic Properties

Grape skin and seeds are rich in antioxidants. Accordingly, their extracts have already shown to exert strong free radical scavenging and chelating activities and inhibit lipid oxidation in various cell models in vitro. Grape phenolic compounds are effective anticancer agents that target epidermal growth factor receptor (EGFR) and its downstream pathways, modifying oestrogen receptor pathways or inhibiting over-expression of cyclooxygenase-2 (COX-2) and prostaglandin E2 receptors [77]. Antioxidant and mitochondria affecting properties of the wine by-products extracts were examined, for example, in human hepatocellular carcinoma (HepG2) cells [72]. All extracts showed protective effects against tert-butyl hydroperoxide (TBH)-induced oxidative stress, although the most pronounced effect was observed with vine shoot extracts that reduced the mitochondrial membrane potential. Further on, Grace Nirmala et al. [78] showed that treatment with wine-making by-products can inhibit the growth of A431 skin cancer cells by inducing cytotoxicity, generating reactive oxygen species followed by loss of mitochondrial membrane potential and induction of apoptosis by exhibiting morphological changes, whereas in normal cells, it was found to be non-toxic. Furthermore, another study showed that leaves’ extracts increase the defence against an excessive production of free radicals and exert a promising anti-proliferative activity against human breast adenocarcinoma [79]. All these studies confirmed that wine by-product extracts rich in bioactive compounds, mainly phenols, could be used as nutraceuticals or as ingredients in functional foods and may offer new opportunities for by-products reutilization.

### 4.2. Tomato Production By-Products

When tomatoes are harvested from the plant, they are washed and sorted. The fruits are then peeled and the tomatoes are again sorted and graded for final processing. During this processing of tomatoes, seeds and peels are obtained as by-products. A huge amount of waste is produced with a significant risk to the environment. Although some of the waste is already being recycled, there is a need for further utilization of tomato waste [80]. Tomato seed waste, which accounts for approximately 60% of the tomato pomace, is a very rich source of nutritional compounds, which includes proteins, amino acids, fatty acids, fibres and functional compounds with important nutraceutical properties [81]. Thus, the superior phytochemical profile of tomato seeds confers to this by-product a high potential.

Due to the high protein content, as well as high nutritional value, tomato seeds have been used as a supplement in animal feeding and as a substitute in bakery products [82]. Currently, the generation of biofuels and the production of enzymes and bioactive compounds are some of the practical applications of tomato waste along with the production of tomato seed oil [83]. Tomato seeds are a rich source of polyphenolic compounds, such as gallic acid, quercetin and trans-cinnamic acid [84]. The phytochemical investigation of tomato seed extract led to the identification of fourteen flavonoids: quercetin, isorhamnetin and kaempferol derivatives. Isorhamnetin-3-O-sophoroside plus kaempferol-3-O-sophoroside and quercetin-3-O-sophoroside were present in the highest concentrations, accounting for 59.1% and 29.2% of the total flavonoids, respectively. Quercetin derivatives contributed approximately 37% of the total flavonoid content [85].

Besides flavonoids, carotenoids are also present in tomato seeds. Carotenoids are widely distributed pigments and are among most chemically and functionally diverse biomolecules on Earth. Carotenoids possess strong antioxidant potential due to their free radicals scavenging ability [86]. Carotenoid profiling of tomato seed oil by using HPLC confirmed the presence of β-carotene and lycopene isoforms [87]. Tomato seeds contain also unsaponifiable compounds, including α- and γ-tocopherols and demethylsterols. Among different demethylsterols present in tomato seeds, delta-5-avenasterol and citrostadienol have high antioxidant activity [88].

Tomato peel and pulp are also rich in bioactive components including flavonoids, phenolic acids, lycopene and β-carotene. Phenolic acids in the peel fraction of tomato are, for example, vanillic acid, caffeic acid, gallic acid and catechin. Tomato bioactive compounds exhibit several biological activities, such as anticancer, antimicrobial, antimutagenic, anti-inflammatory, antineurodegeneration, antiplatelet and cardioprotective properties. Dietary phytochemicals from all types of tomato waste have high potential as functional food and pharmaceutical ingredients [89]. A preview of the biologically active compounds of tomato is presented in Figure 2.

#### Anticancer and Antimutagenic Properties of Tomato By-Products

The preventive and therapeutic anticancer property of tomato is primarily attributed to its carotenoids (i.e., lycopene and β-carotene) and phenolic compounds that modulate several signalling pathways relevant for the cell malignant transformation and/or cancer development. These compounds indeed, primarily target deregulated insulin-like growth factor-1 (IGF-1) () mitogenic pathway [90]. Quercetin-3-O-sophoroside, a major flavonoid found in high amounts in tomato seeds, has been, for example, identified as a potential anticancer compound [91,92,93,94]. Its antitumour mechanisms of action are shown in Figure 3.

Furthermore, several epidemiological studies showed that tomato consumption is associated with a lower risk of cancer [95,96]. It was also found that the tomato glycoalkaloid, α-tomatine, mainly found in the peel, inhibits the growth of colon, liver, breast and gastric cancer cell lines in vitro via caspase-independent signalling pathways [97]. Moreover, Jóźwiak et al. [98] found that tomato seed oil is an excellent source of linoleic acid (48.2%), palmitic acid (17.18%) and oleic acid (9.2%), which have a cytotoxic effect on different cancer cell lines. The cytotoxic effect occurs due to downregulation of cell death regulating factors such as Bcl-2 (B-cell lymphoma 2), whose levels increase due to elevated lipid peroxidation [99]. Additionally, fatty acids in tomato, may generate reactive oxygen species (ROS), which activate caspases and the cleavage of Bid, a pro-apoptotic protein and a death agonist member of the Bcl-2 family. Mitochondrial cytochrome c is consequently released, and apoptosis is activated in cancer cells [100]. These fatty acids also induce cancer cell cycle arrest in G0/G1 and activate the intrinsic apoptosis pathway [101].

Waste generated during tomato fruit processing contains significant amounts of bioactive compounds which could be used as potent antiplatelet, antioxidant, anticancer, antimutagenic, antimicrobial and neuroprotective agents. Therefore, tomato waste has a high potential for development of functional foods as well. However, its health benefits and application in the prevention and treatment of cancers and other diseases should be further investigated. For example, an exaggerated consumption of certain bioactive compounds, often due to inappropriate dosage, may lead to detrimental effects in the body, and data on appropriate applications and dosages are required. Furthermore, there is limited information on the bioavailability and bioaccessibility of several compounds from tomato waste in the human body. Further clinical studies on functional foods with addition of tomato by-product should be performed to evaluate their efficacy and safety [80].

### 4.3. Pomegranate Production By-Products

Pomegranate is also a rich source of beneficial bioactive compounds with confirmed biological effects. However, pomegranate peel, comprising approximately 30–40% of its weight, is usually discarded as a biological waste [102], especially in the production of pomegranate juice. Studies showed that in comparison with other pomegranate fruit parts, the peel contains a high concentration of phenolic compounds with dominating hydrolysable tannins [103,104]. In addition, high-pressure liquid extracts from the pomegranate peel are a valuable source of biologically active compounds for application in the food industry. Pomegranate peel extracts were, accordingly, added in yoghurt samples to increase its antioxidant potential [105], in meat product to improve oxidative stability [106] and in fruits to protect them from mycotoxigenic fungi [107]. Furthermore, the study of [102] showed that ethanol extracts of pomegranate peels exhibit promising antioxidant, antimicrobial and antiproliferative activities in vitro.

Similarly, pomegranate seeds are a rich source of bioactive compounds, i.e., fatty acids, especially unsaturated linoleic, oleic, palmitic, stearic, linolenic, arachidic, and palmitoleic fatty acids [108]. However, it is worth noting that the oil content of the seeds and the fatty acid composition vary according to the growing location, harvest time, fruit genotypes and climatic conditions [109,110]. The seeds are also rich in phospholipids such as lecithin, phosphatidylethanolamine, phosphatidylinositol, phosphatidylcholine and lysophosphatidylethanolamine [108]. Numerous studies have shown that pomegranate seeds have anti-inflammatory, anticancer, antimicrobial and antioxidant effects [111]. The health-promoting effects of phenolic compounds from pomegranate seeds, such as (+)-catechin, (−)-epicatechin, naringin, gallic acid, etc., have also been corroborated by a number of studies [112,113].

#### Anticancer and Antimutagenic Properties

Pomegranate waste might also play a role in cancer drug development or integrative cancer treatment. This is due to a number of observed biological effects of its extracts on tumour cells. For example, the effect of pomegranate seed oil extract on mammary tumours induced by 7,12-dimethyl benz(a)anthracene (DMBA) in rats, showed that treatment with pomegranate seed oil extract had an anti-inflammatory effect and inhibited the expression of the protein (COX-2) around 30–40% [114]. Furthermore, in one study protective effects of pomegranate peel against haematotoxicity, chromosomal aberrations, and genotoxicity was examined. It was confirmed that pomegranate peel alleviated toxicity by scavenging free radicals and preventing DNA damage. It is therefore proven to be a potential preventive product due to its antioxidant and antimutagenic capacities [115]. Other studies reported on the potential of ellagic acid [116] on the induction of the cell cycle arrest and apoptosis, as well as on reduction in prostate cancer and human bladder cancer and leukaemia [117]. The main mechanisms of action proposed for the bioactive pomegranate compounds include inhibition of phase I enzymes or blockage of carcinogenesis, induction of phase II (detoxification) enzymes, the scavenging of DNA-reactive agents, modulation of homeostatic hormones, suppression of hyper-cell proliferation induced by carcinogens, induction of apoptosis, depression of tumour angiogenesis and inhibition of phenotypic expressions of preneoplastic and neoplastic cells [111]. Moreover, Chaves et al. [116] showed that the juice and peel extracts of pomegranate are able to inhibit proliferation, migration and colony formation of prostate cancer cell lines by regulating the mTOR signalling pathway, which controls cell growth and metabolism.

Promising evidence was also gathered on pomegranate seeds potential in protection of other disease as well, including cardiovascular disease and diabetes. Further in vivo and in vitro research and clinical trials are, however, needed to evaluate the relationship between the chemical compositions of pomegranate waste and their mechanisms of action in the treatment of various diseases [113]. These results could have implications for cancer therapy and the food industry related to pomegranate.

### 4.4. Olive Oil Production By-Products

Olive oil is an essential component of the Mediterranean diet and is one of the main food products in Mediterranean countries. However, the production of olive oil and other products derived from olives brings certain burdens to the environment as it generates notable amounts of waste. Many studies have demonstrated a high biological potential of all olive oil by-products [118]. In practice, the application of these by-products is still very low, and their high biological value seems to be neglected. Unfortunately, in the production of olive oil, phenolic compounds from olive fruits are largely retained in the corresponding waste, whereas just a small percentage is transferred to the olive oil [119].

The by-products from *Olea europaea* L. processing industry are already acknowledged as valuable materials for various bioactive molecules extraction, such as polyphenols, anthocyanins, tannins, flavonoids, and dietary fibre (pectin) [120]. The distribution of important compounds in olive oil by-products is presented in Figure 4.

For example, olive leaves are a rich source of phenolic compounds but have not been sufficiently exploited in health-promoting application. Accordingly, olive leaves from different cultivars may be easily exploited for preparation of functional olive leaf infusions rich in bioactive compounds [102]. Furthermore, several studies have shown that extracts from olive leaves can significantly reduce the risk of cardiovascular disease, due to their anti-atherosclerotic, hypotensive, antioxidant, anti-inflammatory and hypocholesterolaemic effects [122]. Moreover, bioactive compounds have strong antimicrobial and neuroprotective activities as well [123]. The exact phenolic composition of olive leaves is usually assessed by use of HPLC, but a complete structural characterization of phenolic compounds may be performed by mass spectrometry (MS)-based methods, which provide for higher selectivity and sensitivity [124].

Further on, pomace is a solid by-product obtained upon mechanical extraction of virgin olive oil. The composition of the pomace is variable and depends on the extraction system, olive variety and the degree of ripeness of the fruit. This by-product is currently seen more as a problem instead as a valuable bioactive compounds resource. This is probably due to large quantities of polyphenols and stone (or kernel) remaining in the pomace at the end of the production process. A pomace without stone may be obtained by recently introduced multi-phase decanters. The chemical characterization of this new olive oil by-product called “pâté”, consisting only of the pulp and vegetation water, without traces of stone, contains hydroxytyrosol, verbascoside and oleuropein aglycon derivatives in high levels [125]. In addition, debittering of the pâté for increased edibility was also investigated [126]. Sequential filtrations of the pâté showed to be faster and more efficient in phenolic degradation and therefore a better method to make the pâté suitable as an ingredient for food preparation. The authors also reported that dry extracts of pâté had reduced expression of those compounds that might be relevant antitumor effects and modulation of mitochondrial activity. Further studies are needed to better explore the pâté therapeutic potential. However, all studies performed so far confirmed that the olive oil by-product pâté has a high reusage potential for production of functional food or food supplements.

In summary, many health benefits are associated with consumption of polyphenols from olive oil. These polyphenols are present at a higher abundancy in olive by-products. Recent scientific research has shown that valuable compounds can be recovered from olive by-products by use of green extraction technologies. These compounds and reused for food, pharmaceutical and cosmetic applications, contributing to a circular economy [121].

#### Anticancer and Antimutagenic Properties

The results of in vitro and in vivo studies reported that polyphenols from olive oil or olive oil by-products (specifically phenylethanoids) have a chemopreventive and anticancer potential. A combination of these compounds with anticancer drugs may thus been evaluated as an integrative strategy for cancer treatment. Still, apart from positive data from preclinical studies showing beneficial effects of olive oil polyphenols alone or in the combination with anticancer drugs, their efficacy remains to be proven in humans [127].

It is known, for example, that oleuropein aglycone, found in olive fruit, has antihyperglycemic, neuroprotective, anti-inflammatory, antioxidant and other properties [128]. Several studies, accordingly, investigated its influence on cancer cells and showed promising results. In a study conducted by Mazzei et al. [129] oleuropein aglycone was tested as an antiproliferative drug toward breast cancer cells. The results demonstrated that it caused a selective inhibition of cell growths. It induced G0/G1 cell cycle arrest, enhanced the cyclin-dependent kinase (CDK) inhibitor p21Cip/WAF1, p27, p53 expression, and decreased cyclin D/E expression at both mRNA and protein levels. This study has shown that oleuropein aglycone may be a potential agent for the treatment of breast cancer, especially to overcome resistance to hormone therapies. Luteolin and apigenin from olive by-products have also been identified as compounds responsible for antitumor effects due to their ability to reduce oxidative damage and modulate NF-κB-mediated inflammatory responses and tumour progression-related pathways [130].

### 4.5. Citrus Fruit Production By-Products

Citrus fruits are the highest yield crop worldwide, and about 34% of harvested citrus fruits are used to produce juices [131]. The most important citrus fruits are sweet orange (*C. sinensis*), bitter orange (*C. aurantium*), lime (*C. aurantifolia*), lemon (*C. limon*), bergamot (*C. bergamia)*, grape fruit (*C. paradisi*) and mandarin orange (*C. reticulata*) [132]. Peels are the main by-product of citrus juices and other products derived from citrus fruits, which are mostly wasted today [131]. However, dry citrus peels are rich in pectin, cellulose, and hemicellulose and can be used as a fermentation substrate, processed into cattle feed, or used to extract essential oils and d-limonene [133]. In addition, citrus waste is also used as a source of fibres for food fortification, or polyphenols as natural antioxidants, or even as adsorbents to remove dyes, or for microbial production of pectin [134]. Besides that, citrus peels have been developed as a promising renewable resource for the production of biofuels. Extraction and drying can maximize the nutrition and bioactivity of citrus peels and prolong its storage life. There is no doubt that citrus peels are a valuable source of pharmacologically active components with a confirmed content of flavonoids, limonoids, alkaloids, essential oil and pectin. Most importantly, the health-promoting relevant effects of citrus peels are most likely due to flavonoids [135].

The exploitation of agro-industrial waste using biotechnological methods has attracted great interest in recent years. The purpose of the process is to utilise organic compounds that can serve as a source of carbon and energy for the growth of microorganisms or to produce bioactive compounds. It was found that orange peel waste has a high concentration of polyphenols, being 28% condensates, 27% ellagitannins, 25% flavonoids and 20% gallotannins. The major polyphenolic compounds were catechin, ellagic acid and quercetin [136].

Gómez-Mejía et al. [137] reported a fast and sustainable method that combines solid–liquid extraction based on ethanolic aqueous solution, chromatographic analysis and chemometrics to extract and quantify polyphenols from citrus peels. The authors suggested that clementine peel could be a good source of hesperidin, orange peel could provide higher contents of rutin, while lemon peels can be used due to high amounts of naringin. Based on this research, citrus peel waste might be reused for development of functional foods, cosmetics, or preventive therapies for some diseases.

Finally, citrus peels contain a rich array of bioactive components that might find a way into innovative and functional foods production/reusage. Thus, the health-promoting effects of citrus peels extracts corroborated by in vitro and in vivo studies, require further investigation on in vivo models. Moreover, an improved exploitation and understanding of citrus peels content, genetic studies of diverse species should be pursued on to uncover compositional variation and functional properties [135].

#### Anticarcinogenic and Antimutagenic Properties

Citrus fruits and leaves are traditionally used for anticancer applications in India and other countries [138]. In the Mediterranean countries, the usage of *C. medica* and *C. aurantium* as anticancer agents is well known [139]. The anticancer effect of essential oil of *C. aurantifolia*, *C. paradisi*, *C. deliciosa*, *C. aurantium*, *C. limon*, *C. jambhiri* and *C. pyriformis* has been investigated on various cancer cell lines. For example, ellagic acid, commonly found in citrus fruits, has been shown to inhibit tumour growth without causing cardiotoxicity in mice [140]. In addition, flavonoids, limonoids, vitamin C, and β-carotene have also been identified as the major antioxidants of citrus species responsible for reduction in cancer risk. Citrus is one of the major fruit crops globally, especially in the Mediterranean region, and its peel is the main residue during citrus industrial processing. New scientific findings will provide deeper insight into how citrus peels can be used to develop useful functional foods, medicines, and biofuels [134].

## 5. Anticarcinogenic Properties of Specific Compounds Found in Mediterranean Food and Its Industrial Waste

Abnormal cell growth inside a living organism that has potential to invade and metastasize to other parts of the body is usually acknowledged as cancer. A change in gene or genes’ expression that regulate normal body functions occurs at a very beginning of malignant cell transformation. Cancer is currently one of the leading causes of death in the world [141]. Due to this and due to limitations of conventional treatments, biotechnology companies are intensively working on finding new solutions to this problem [142]. Until now, eight malignant cells hallmarks acknowledged in the basis of cancer pathogenesis and these are the acquired capabilities for sustaining proliferative signalling, activating invasion and metastasis, evading growth suppressors, enabling replicative immortality resisting cell death, inducing/accessing vasculature, reprogramming cellular metabolism and avoiding immune destruction. Recently, additional new factors have also been considered core hallmarks of cancer. These are nonmutational epigenetic reprogramming, unlocking phenotypic plasticity, polymorphic microbiomes and senescent cells, and have full potential for understanding the complexities, mechanisms, and manifestations of the disease [143].

In particular, nonmutational epigenetic regulation of gene expression of cells is well established as the central mechanism mediating embryonic development, organogenesis and differentiation [144]. Since epigenetic mutations are reversible in contrast to genetic defects, and since epigenetic deregulation of normal cells is one of cancer hallmarks, the process is growingly acknowledged as an important target for cancer therapy. One possibility is to apply already known compounds from plants or food by-products to counteract unwanted epigenetic changes in cancer cells. For example, polyphenols, such as resveratrol and catechin, may be used as chemopreventive agents that can reverse epigenetic signatures of cancer cells. Some polyphenols are currently evaluated for their ability to reverse adverse epigenetic marks in cancer (stem) cells to attenuate tumorigenesis-progression, prevent metastasis or sensitize for drug sensitivity [145]. Furthermore, increasing evidence points unlocking of phenotypic plasticity typical of cancer cells as an important therapeutic target [146]. For example, a study showed beneficial effects of quercetin in cardiovascular diseases [147]. This may be correlated to cancer as vascular smooth muscle cells (VSMCs) undergo a unique phenotypic modulation during development as well as in response of blood vessel remodelling or injury. This phenotypic plasticity of VSMCs contributes to vascular disease by allowing for differentiation to inappropriate lineages [148]. Accordingly, quercetin may be evaluated as a potential pharmacological tool for treatment disease underlined by phenotypic plasticity [147]. Interestingly, in various cancer types, variability of the microbiome composition between individuals in a studied population might have a significant impact on the final cancer phenotype in an individual patient. Specific microorganisms, indeed, have either deleterious or protective effects on cancer development, and they impact tumour progression and therapy [149]. For example, dietary and salivary proteins influence the composition of the oral microbiome, where, for example, effects of daily exposure to a cranberry polyphenol oral rinse effected the salivary proteins and oral microbiota content and abundance, which point to oral health and gut health in the basis of the health status [150].

Additionally, cellular senescence is a protective mechanism against neoplasia [151]. Targeting senescence is thus, an emerging anti-cancer strategy where polyphenols have a significant role. One study has shown, accordingly, that polyphenols inhibit development of cancerous microenvironment from the senescent cells. Once tumorigenesis occurs, polyphenols inhibit cancer growth by oncogene-, oxidative stress-, DNA damage response (DDR)-, and endoplasmic reticulum (ER) stress-induced cancer cell senescence. Use of polyphenols for this kind of therapy will be a key consideration in the future [152].

Considering all of the above, a polyphenol-assisted nanoprecipitation approach for engineering of functional nanoparticles is gaining more significance due to increased polyphenol availability to bodily cells. By optimization of the weight ratio of carrier materials and polyphenols, type of polyphenols and type of nanoparticles, the polyphenol nanoparticles may be used as a nanotherapy tool for targeted treatment of various diseases, including cancer [153].

Indeed, chemoprevention based on the use of biological, natural, or synthetic agents should be seriously considered in cancer management. This field is categorized into primary, secondary and tertiary chemoprevention, which includes a whole array of patients, namely, from a population with no cancer, low and high risk population to the population that already has malignant lesions and cancer [142]. In chemoprevention, bioactive compounds found in various by-products may easily be exploited in cheap management set-ups. Many bioactive compounds indeed, interfere with tumour development and may act as protective agents [154,155]. Table 1 gives an overview of bioactive compounds with antitumour effects that can be found in the Mediterranean food and its industrial waste.

For example, majority of plant polyphenols are capable of preventing the malignant transformation of cells and therefore decrease the risk of developing cancer. Moreover, a number of studies corroborate the essential role of polyphenolic compounds from fruits, vegetables or herbs, especially found in the Mediterranean area, in cancer protection through epigenetic modifications regulation [176]. In Figure 5, a schematic review of flavonoids and some of their main action pathways is given as an example of their mechanism of action in vivo. Through a series of molecular events, flavonoids enter the cells and, by activating some molecules involved in the cell proliferation and cell death, such as the tumour suppressor p53, cause further cascade events which ultimately lead to apoptosis of a cell. This underlies the anti-proliferative effect. Another well-known pathway involved in malignant progression is Akt/FAK/Ras/PI3K pathway, that underlies the anti-metastatic effect of flavonoids as well. Another significant pathway in tumour cells is the mitogen-activated protein kinase (MAPK) pathway, which may act as an anti-inflammatory. These are just some crucial pathways targeted by flavonoids, and other signalling pathways in the cell may be triggered by flavonoids as well.

For example, it has been shown that naringenin can inhibit proliferation, invasion, and migration of cancer cells and induce apoptosis in patients with lung cancer, which may be related to its inhibition of the Akt signalling pathway [159]. Another study has shown that naringenin could potentially suppress tissue plasminogen activator (TPA)-induced cancers by down-regulating multiple signalling pathways such as extracellular signal-regulated kinase (ERK) and c-Jun N-terminal kinases (JNK) signalling pathway and NF-kappaB signalling pathway [158]. Apigenin is another flavonoid that possesses strong anti-cancerogenic properties. Treatment with apigenin significantly suppressed the aggressive type of breast cancer cells, the HER2-positive breast cancer type, by inhibiting the signal transducer and activator of transcription 3 (STAT3) signalling pathway [162]. Furthermore, in prostate cancer, apigenin treatment induced cell cycle arrest and apoptosis, thus significantly inhibiting tumour growth in mice by inhibiting Histone deacetylases (HDACs), especially HDAC1 and HDAC3 expression in prostate cancer cells [162]. The protective anti-cancer effect is also well documented for the flavonoid quercetin. Treatment with quercetin effectively inhibited proliferation and induced apoptosis of HCC (hepatocellular carcinoma) cells through up-regulation of Bad and Bax and therefore reduced aggressive tumour growth in mice models [177]. As it interferes with Bax regulation, its role in MCF-7 breast cancer activation quercetin inhibits insulin receptor signalling and therefore impairs the proliferation of breast cancer cells [178]. In another study, quercetin inhibited breast cancer cell growth and migration by interfering with the VEGFR2-mediated pathway, which is crucial for cancer angiogenesis and development [167].

Antiproliferative and pro-apoptotic effects in a variety of cancer cell lines was also shown for phenolic acids. Ellagic acid showed inhibitory effects on MCF-7 breast cancer cells by cell cycle arrest via the TGF-Smads signalling pathway [168]. On the other hand, considering prostate cancer cells, a study by Pitchakarn et al. [179] suggests that a certain dose of ellagic acid significantly suppressed the cell motility and invasion by down-regulation of MPPs. Furthermore, administration of gallic acid was shown to reduce cancer cell viability in colon and prostate tumours, due to ROS-dependent pro-apoptotic effects [180]. Moreover, it is reported that treatment with gallic acid selectively inhibited the growth of liver cancer cells by acting via mitochondria-mediated apoptotic pathways [181].

In a context of the Mediterranean diet and its by-products, three compounds may be particularly emphasized: resveratrol, hydroxytyrosol and oleuropein. Stilbenes, including its main representative resveratrol, can differentially influence DNA methylation in tumour suppressor genes and oncogenes. It has also been found that the SALL3 gene, is upregulated by stilbenes, which down-regulates DNMT3A (DNA (cytosine-5)-methyltransferase 3A), binding to the promoters of tumour suppressor genes such as SEMA3A [182]. Furthermore, stilbenes influence Nuclear Factor 1C protein, a tumour suppressor that localizes heavily on the SEMA3A promoter. A combined treatment of both resveratrol and pterostilbene down-regulated SIRT1 and HDAC III, which are responsible for leading DNMTs to the hypermethylation of promoters and silencing of TSGs. Furthermore, it has been shown that stilbenes can successfully down-regulate oncogenic activity of PRMT5 and EZH2 via reducing silencing histone [183]. Another study found that the application of stilbenes, such as resveratrol and pterostilbene, can silence tumour suppressor genes in breast cancer tissues via acetylated transcription factor STAT3 [172]. In this context, STAT3 phosphorylation also plays an important role in tumorigenesis, and it is proven that stilbenes can successfully prevent this process [173]. Although several promising treatments using stilbenes have been discovered, further research on novel epigenetic reactions and targets is required to further understand their anti-cancerogenic effects. On the other hand, it has been documented that stilbenes enhance the cytotoxic effects of DNA damaging agent and irradiation therapy by targeting SIRT1-regulated γ -H2AX expression in cancerogenic cells, with no overall effects on control breast epithelial cells. All these findings taken together imply a promising future for these dietary compounds [184,185].

Resveratrol provides a wide range of therapeutic and preventive options against a variety of cancer types, including breast cancer, prostate cancer, colorectal cancer, lung cancer, thyroid cancer and melanoma. It could be used as an agent in different anticancer therapies. Due to its natural origin, low cost, and safety compared to other drugs, it may be a useful complementary medicine for the treatment and prevention of various cancers, although further studies on this natural compound are needed [186]. Figure 5 presents a review of a schematic diagram that represents the potential mechanism/s underlying the anticancer effects of resveratrol, including the initiation, promotion and progression phase of cells, as well as its potential use in chemoprevention and therapeutic effect.

Oleuropein is a polyphenolic compound in olive oils and the leaves of the olive tree that received considerable attention in the scientific community due to promising health effects. Together with hydroxytyrosol, it has a powerful antioxidant activity, which may be responsible for previously observed antioxidant, anti-inflammatory, and disease-fighting properties of olive oil. Many studies confirmed anticancerogenic effect of oleuropein, which was observed in breast adenocarcinoma, melanoma, urinary bladder carcinoma, colorectal adenocarcinoma, prostate cancer, lung cancer and glioma [187]. It was found that oleuropein displays overall anticancer activity by its antiproliferative and proapoptotic effects. One study has shown that oleuropein induces apoptosis in breast cancer cells acting via p53-dependent pathway by regulating Bax and Bcl2 genes [175]. Definitely, oleuropein has a potential to be prescribed as therapeutical agent in future for breast cancer patients [187,188,189]. Finally, the strong antioxidant effect of oleuropein should be highlighted, as it is responsible for protecting against genetic damage that may lead to oncogenesis [190]. Degradation products of oleuropein, hydroxytyrosol is a phenolic compound found in olive fruits and leaves. It is considered to be one of the most powerful antioxidant compounds, with different health benefits including anticancerogenic properties [191]. It has been demonstrated that despite its relatively low concentrations in olive oil, hydroxytyrosol reduces epidermal growth factor (EGFR) level by promoting its degradation. EGFR is one of the main receptors that trigger colon cancerogenesis as it regulates the angiogenesis, invasion of cancer cells, proliferation and apoptosis. It was also found that HXT is an effective cytotoxic agent in breast cancer, due to inhibition of G0/G1 phase [174].

## 6. Future Prospective

It is well documented that food may strongly modulate innate and adaptive, immune responses. This evidence underlies development of a new discipline known as ‘immunonutrition’. As tumour growth is highly dependent on the host immune system status, efforts were put in both, natural-based and pharmaco-based strategies for modification of the immune system in the tumour microenvironment [192]. Accordingly, a number of studies have been carried out to determine the effects of individual food components on the immune response through the in vitro or in vivo experiments in humans and animal models. Obtained results are certainly encouraging. It was shown, for example, that tested nutraceuticals acted on the immune system differently and this effect varied according on the type of tumour and target immune cells. In addition, they could even have an antagonistic or synergistic effect in patients. Therefore, it is important to consider the patient intervariability and eventual cumulative effects of nutraceutical intake as well. However, the majority of phytochemicals have a low absorption and biodistribution rate, making them safe for consumption. In addition, they are metabolized swiftly in the body and excreted from the human body as well. The poor bioavailability on the other side, poses challenges in delivery of an effective dosage to the target cancer cells and into the tumour microenvironment. Nanotechnologies can also help improve the tissue delivery [192]. Nanoencapsulation of these polyphenols could prolong circulation improve localization, enhance efficacy and reduce the chances of multidrug resistance [193]. Still, the toxicology parameters of nanocarriers should be carefully designed and studied for such applications.

Additionally, if by-products are re-used, it is necessary to control the pesticide or other contaminants content in the new products. By-products are indeed a source of bioactive compounds, but may contain significant amounts of pesticides and contaminants as well. Pesticides are, for example, directly linked to cancer occurrence. One study showed, for example, that up to 39 new estimated cases of cancer could occur annually due to pesticides residues in fruits and vegetables, which is not at all negligible. The numbers are not higher due to being counterbalanced by the protective effect of fruits and vegetables against various types of cancer [194]. It is stated that between 3374 and 4407 annual cancer cases were prevented by reported consumption of fruits and vegetables [195,196] On the other hand, very strict rules when it comes to usage of pesticides should be discussed and revised, and they should include the real necessity for dosage and the type of used pesticide. A quest for alternative pest management methods for crops should indeed be the primary goal of the food industry, where the impact on human exposure and risk should be considered as a primary goal [196].

Resveratrol may be particularly considered as a promising anti-cancer agent. However, resveratrol is very sensitive to degradation by light, temperature, oxygen exposure and oxidative enzymes, and nanocapsulation might solve some of these barriers to its proper preservation within different products [197]. For example, enhanced antiproliferative efficacy, enhanced bioavailability and no-observable toxicity was reported for resveratrol-loaded gelatin nanoparticles on non-small-cell lung cancer cells [198]. Furthermore, considerable reduction in cell number and colony-forming capacity of colon cancer cells along with increased apoptosis and reduced ROS by PEG-PLA NPs was observed in resveratrol treatment in another study [199]. Resveratrol–bovine serum albumin NPs were responsible for the induction of cell death pathway in human ovarian cancer cells as well [200]. Encapsulation of resveratrol with casein NPs significantly increases oral bioavailability of resveratrol. Recently, another study reported the antihepatoma effect of nanogold-loaded resveratrol that induces the inhibition of cell proliferation and promotion of apoptosis in HepG2 cells without any observable toxicity. The mechanism of action of this nanoformulation was reported to be the downregulation of procaspase-3, procaspase-9, PI3K and Akt and the upregulation of caspase-8 and Bax [193].

Besides traditional knowledge on malignant targets, new targets may be studied as an effort for future technologies applications. For example, ubiquitin–proteasome pathway (UPP) is essential for the maintenance of cellular homeostasis, because it regulates key processes including cell cycle or DNA damage response. This is why UPP may be considered as an interesting target for natural products-based anti-cancer therapy. Several polyphenols have, for example, already been reported to exert proteasome-inhibitory activity; here, the accumulation of ubiquitinated proteins was achieved along with the suppression of proliferation and apoptosis induction in cancer cell lines [201]. However, the potential use of UPP as an polyphenols anti-cancer target is still at very early stage of research. Further clinical studies are needed to determine potential efficacy of natural bioactive compounds as inhibitors of UPP in the context of cancer therapy.

Recent clinical studies on cancer therapy are conducted with common naturally derived polyphenols such as resveratrol, curcumin, and quercetin. Nowadays, most of these clinical studies are still in progress. Research of cancer therapies using various polyphenol families, particularly flavonoids, has contributed to the development of natural remedies that are less aggressive than conventional anticancer drugs. In fact, various studies have shown that polyphenols could be used as chemotherapy adjuvant agents in cancer therapies [202].

Re-usage of by-products for production of new food has some limitations. These new products must comply with all quality and safety requirements of the food regulations for a given country. Another potential limitation is a higher content of antinutritional molecules in such reused products. Some of these are condensed tannins, saponins, trypsin inhibitors and others. These compounds are generated by the secondary metabolism of the plants as a protection from predators or environmental threats, and are found at higher concentrations in the food waste after primary usage of nutrient-rich food parts. This may lead to a problem of interference of such molecules with the bioavailability and digestibility through binding to other nutritional compounds such as proteins, carbohydrates and minerals. Limitations will need to be solved for the best possible benefits that might be withdrawn from by-products. Overall, the future challenge will be to find nutraceutical formulations with improved bioavailability that can be safely used in combination with traditional antitumor therapies without any negative interference with other food nutrients [192].

## 7. Conclusions

The Mediterranean diet as a dietary pattern has many beneficial effects on human health and can have a role in prevention of non-communicable diseases. In particular, bioactive compounds of the Mediterranean diet act as cell-protective factors due to their antioxidative and anti-inflammatory properties. A number of observed molecular mechanisms of naturally derived bioactives from plants action are directed towards hallmarks of cancer. The bioactive compounds of the Mediterranean diet are particularly enriched in by-products of the Mediterranean food industry and may be readily exploited in different food, cosmetics and pharmacological applications. However, further studies are needed to develop effective extraction methods for these compounds from by-products and to deepen the understanding of their effects on human health. An interesting field of application is also the development of formulations with increased bioavailability.

## Figures and Tables

**Figure 1 molecules-27-08655-f001:**
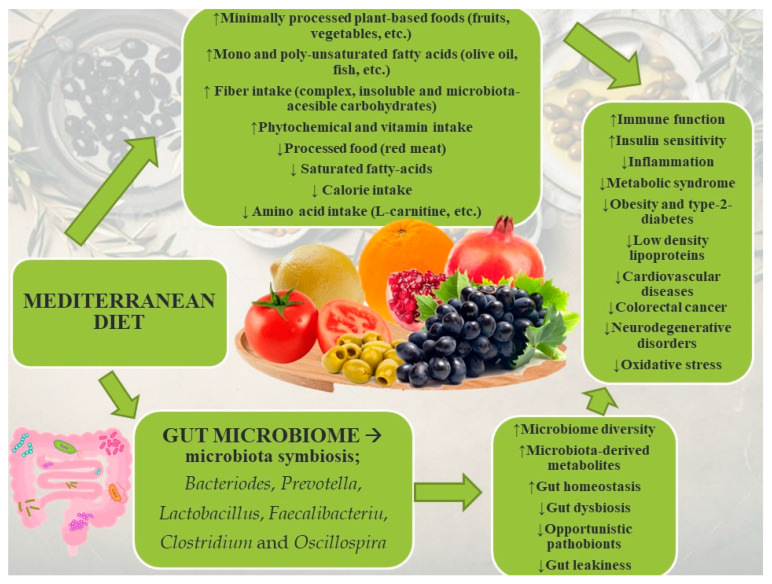
Characteristics of the Mediterranean diet and associated health benefits. The positive effects of the Mediterranean diet are also connected with specific effects on the microbiota composition.

**Figure 2 molecules-27-08655-f002:**
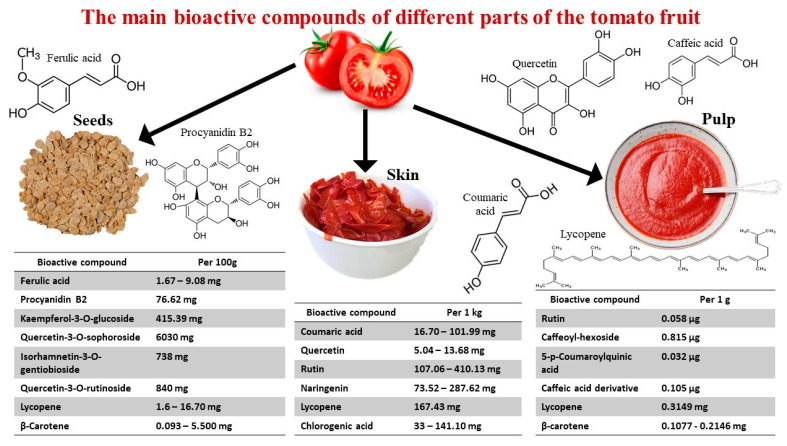
Main bioactive compounds of different tomato fruit parts, namely, seeds, skin/peel and pulp. Quantitative data adapted from Kumar et al. [80].

**Figure 3 molecules-27-08655-f003:**
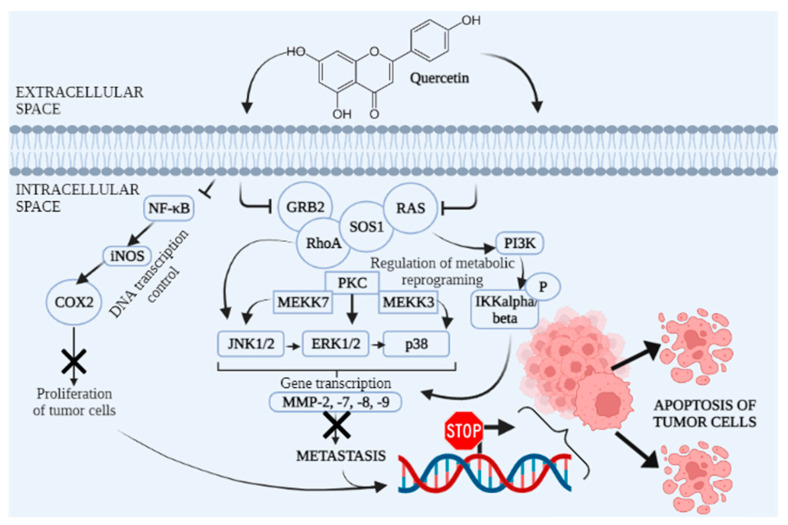
Scheme of quercetin antitumor mechanisms of action. NF-κB (Nuclear factor kappa-light-chain-enhancer of activated B cells)-iNOS (Inducible nitric oxide synthase)-COX2 (Cyclooxygenase 2) is an inflammatory signalling pathway which is blocked by quercetin induction in the tumour cell. It results in blockade of proliferation of tumour cells. Moreover, quercetin inhibits interaction of GRB (Growth factor receptor-bound protein), RhoA (Ras homolog family member A), SOS1 (Son of Sevenless 1) and RAS (Rat sarcoma virus), which prevents further metabolic reprograming. That includes inhibition of other metabolic molecules; PKC (Protein kinase C), MEKK-3, -7 (Mitogen-activated protein/ERK kinase -3, -7), JNK1/2 (c-Jun N-Terminal Protein Kinase 1/2), ERK1/2 (Extracellular signal-regulated kinase 1/2), p38 (Mitogen-activated protein kinase), PI3K (Phosphoinositide 3-kinase), IKKalpha/beta (Inhibitor of nuclear factor kappa-B kinase subunit beta) and, ultimately, further genetic transcription through MMP-2, -7, -8, and -9 (Matrix metalloproteinases). Finally, these events lead to tumour cell apoptosis.

**Figure 4 molecules-27-08655-f004:**
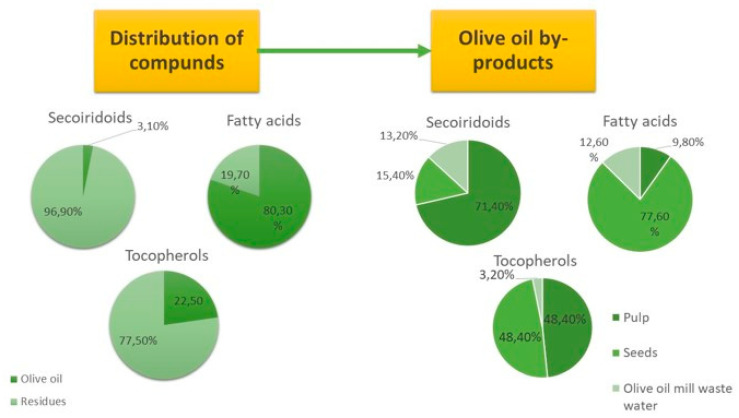
Content of important bioactive compounds of the olive oil and olive oil by-products that can be further used in food industry and medical application [121].

**Figure 5 molecules-27-08655-f005:**
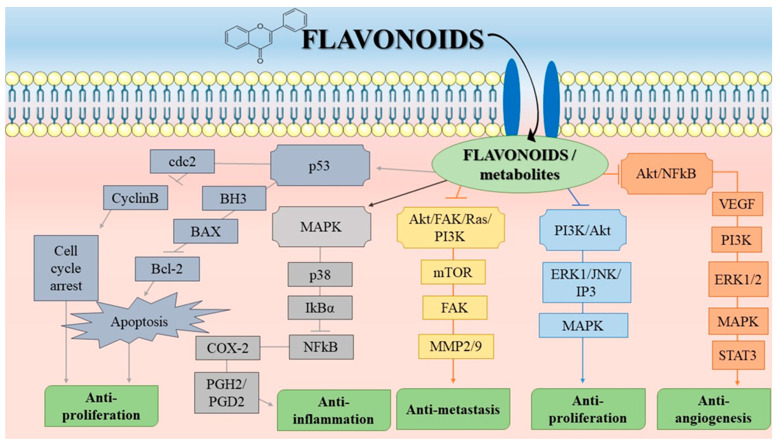
Schematic overview of the most common signalling pathways through which flavonoids act on malignant processes in the cells. The final outcomes on the molecular level of cell organization include anti-proliferation, anti-inflammation, anti-metastasis and anti-angiogenesis effects. By acting upon p53 (cellular tumour antigen) molecule, which further acts on inhibiting cdc2 (Cyclin-dependent kinase) and CyclinB and therefore causes cell cycle arrest with anti-proliferation as an end result. Furthermore, p53 has an impact on blockading BH3 (borane trihydridoboron), BAX (BCL2 Associated X, Apoptosis Regulator) and Bcl-2 (B-cell lymphoma 2) molecules which leads to apoptosis and also causes inhibition of proliferation. Another significant molecule is MAPK (Mitogen-activated protein kinases), which further blocks the p38 (tumour suppressor), IKBalpha (nuclear factor of kappa light polypeptide gene enhancer in B-cells inhibitor, alpha), NFkappaB (Nuclear factor kappa-light-chain-enhancer of activated B cells), COX2 (Cyclooxygenase) and PGH2/PGD2 (Prostaglandin H2/D2) pathway, with a final result of inhibition of inflammation. Another pathway on which flavonoids act upon is Akt (Protein kinase B, PKB)/FAK (Focal adhesion kinase)/Ras (Rat sarcoma virus protein)/PI3K (Phosphoinositide 3-kinases) pathway, through mTOR (mammalian target of rapamycin), FAK and MMP2/9 (matrix metallopeptidase 2/9) molecules. Blockade of Akt/FAK/Ras/PI3K pathway causes inhibition of metastasis. PI3K/Akt molecules are also significant in anti-proliferation module of flavonoid action, through another group of molecules, MAPK and ERK1 (Extracellular signal-regulated kinases)/JNK (c-Jun N-terminal kinases)/IP3 (Inositol trisphosphate). Finally, anti-angiogenesis properties that arise from flavonoid action is through Akt/NFkappaB pathway, which includes VEGF (Vascular endothelial growth factor), PI3K, ERK1/2, MAPK and STAT3 (Signal transducer and activator of transcription 3) molecules.

**Table 1 molecules-27-08655-t001:** Summarized list of bioactive compounds with antitumour properties along with their origin, target cancer type and signalling pathways.

Anticarcinogenic Compound	Cancer That Impacts	Action Pathways	Source	Group of Compounds	Reference
Lycopene	prostate cancer, breast cancer and cervical cancer	NF-kappaB signal transduction, angiogenic effects, inhibition of cancer promotion signalling Wnt-TCF signalling PI3K-Akt and ERK/p38 signalling pathways	tomatoes but also in big variety of other fruits and vegetables	Carotenes	[156,157]
Naringenin	lung cancer, TPA-induced cancer, breast cancer	Akt signalling pathway ERK and JNK signalling pathway NF-kappaB signalling pathway	mostly found in some edible fruits, such as citrus species and tomatoes.	Polyphenols Flavonoids Flavanones	[158,159]
Hesperitin	gastric cancer, breast cancer, cervical cancer, prostate cancer	intracellular ROS accumulation ASK1/JNK pathway NF-kappaB pathways	lemons and sweet oranges as well as in some other fruits and vegetables	Polyphenols Flavonoids Flavanones	[160,161]
Apigenin	lung cancer, colorectal cancer, prostate cancer, breast cancer	STAT3 signalling pathway DNA damage and apoptosis	plant-based food such as oranges, onions, parsley, tea and wheat sprouts	Polyphenols Flavonoids Flavones	[162]
Luteolin	lung cancer, colon cancer, breast cancer, prostate cancer	NF-kappaB signalling pathway JNK activation pathway Suppression of the expression of integrin_1 and FAK down-regulation expression of prostate-specific antigen IGF-1-mediated PI3K/Akt and ERK1/2 pathways	abundant in Mediterranean spice plants (sage, oregano and thyme) and artichoke	Polyphenols Flavonoids Flavones	[163,164,165]
Quercetin	lung cancer, gastric cancer, colon cancer, liver cancer, breast cancer, prostate cancer	VEGFR2-mediated pathway Bax regulation insulin receptor signalling AMPK-mediated signalling pathway	mostly in onions, grapes, berries, cherries, broccoli, and citrus fruits	Polyphenols Flavonoids Flavonols	[166,167]
Ellagic acid	liver cancer, breast cancer, prostate cancer	TGF-_/Smads signalling pathway down-regulation of MPPs inhibition of cell proliferation	predominantly found in some nuts as chestnuts, walnuts, pecans, furthermore cranberries, raspberries, strawberries, and grapes, as well as distilled beverages like wine. It is also found in pomegranates	Polyphenols Flavonoids Isoflavones	[168]
Gallic acid	gastric cancer, colon, prostate cancer, cervical cancer	up-regulation RhoB and down-regulation AKT/small GTPase signals NF-kappaB signalling ROS-dependent pro-apoptotic effects down-regulation of MMP-2 and MMP-9 mitochondria-mediated apoptotic pathways	blackberries, raspberries and other forest fruits, as well as in the products such as wine and chocolate	Polyphenols Flavonoids Isoflavones	[169]
Resveratrol	breast cancer, lung cancer, colon cancer, liver cancer, prostate cancer, adenocarcinoma, skin cancer, thyroid cancer	cancer cell G1 phase arrest and by promotion generation of ROS induction DNA damage and apoptosis suppression of expression of KRas down-regulation of the HER-2/neu gene expression SP-1 signalling pathway Suppression of the mRNA level of androgen-responsive glandular kallikrein 11 induction of cytochrome c release, the expression of Bax, p53, and Apaf-1, and the inhibition of Bcl-2 to lower expression levels of TGF-1 and augmented expression levels of E-cadherin,	predominantly found in berries, grapes and wine	Polyphenols Stilbenes	[170,171]
Pterostilbene	colon cancer, breast cancer, prostate cancer	activation of AMPK in p53 gen Bax activation and its over-expression Inhibition of cancer cell growth and induction apoptosis, as well as the S phase cell cycle arrest	mainly found in blueberries	Polyphenols Stilbenes	[172,173]
Piceatannol	Prostate cancer	down-regulation of mTOR up-regulation of miR-129 and, thus, down-regulation Bcl-2, which is a known target of miR-129	in variety of Mediterranean foods such as grapes, berries, passion fruit, etc.	Polyphenols Stilbenes	[172,173]
Hydroxytyrosol	Breast cancer, colon cancer	inhibition of G0/G1 phase reduction in epidermal growth factor (EGFR) level by promotion its degradation	olive tree and its leaves as a by-product obtained from the manufacturing of olive oil	Polyphenols Phenylethane	[174]
Oleouropein	breast adenocarcinoma, melanoma urinary bladder carcinoma, colorectal adenocarcinoma, prostate cancer, lung cancer, glioma	p53 dependent pathway by regulating Bax and Bcl2 genes	enriched in olive oil and leaves of the olive tree	Polyphenols Phenylethane	[175]

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
