# Peer review of "Mediterranean Food Industry By-Products as a Novel Source of Phytochemicals with a Promising Role in Cancer Prevention"

_molecules, 2022, doi:10.3390/molecules27248655_

Round 1

Reviewer 1 Report

This paper entitled “Mediterranean food industry by-products as a novel source of phytochemicals with a promising role in cancer prevention” authored by Andrea Agaj et al, presents by-products from the Mediterranean food industry as a source of phytochemicals and gives details on mechanisms in a molecular level that are related to anticancer activities

The review is very interesting. The authors present data in a well-structured and comprehensive way. Informative table and figures and up-to-date cite references have been used. The conclusions are useful for future works.

However, I do have two minor comments to improve the manuscript before it can be accepted for publication:

Minor Comments:

1.      I suggest that table 1 should be embedded in the text

2.      Authors should correct figure 1 in order to make visible the resulting effect on Neurodegenerative disorders and Oxidative stress

Author Response

Responses to the Reviewer #1comments

This paper entitled “Mediterranean food industry by-products as a novel source of phytochemicals with a promising role in cancer prevention” authored by Andrea Agaj et al, presents by-products from the Mediterranean food industry as a source of phytochemicals and gives details on mechanisms in a molecular level that are related to anticancer activities.

The review is very interesting. The authors present data in a well-structured and comprehensive way. Informative table and figures and up-to-date cite references have been used. The conclusions are useful for future works.

However, I do have two minor comments to improve the manuscript before it can be accepted for publication:

Reply: We would like to thank the reviewer for helpful remarks and suggestions. All the comments are now carefully and thoroughly implemented. We hope that now the new version of the manuscript will be recommended for publication. The detailed answers are below.

Minor Comments:

  1. I suggest that table 1 should be embedded in the text

 Reply: The table 1 is embedded in the text, thus facilitating the reading sequence.

  1. Authors should correct figure 1 in order to make visible the resulting effect on Neurodegenerative disorders and Oxidative stress

Reply: Thank you again for bringing our attention to the pointed mistake, indeed, the effect on Neurodegenerative disorders and Oxidative stress was not given explicitly in the figure. We have now included the revised Figure 1. In our manuscript.

Reviewer 2 Report

This review article provides an overview about the Mediterranean food industry by-products as a novel source of phytochemicals with a promising role in cancer prevention. The concept is very novel and demanding in context to finding an alternative solution for natural compounds in disease treatment specifically for cancer. Authors described the concept in very well manner, but still manuscript requires some corrections. My suggestions are:

1. Authors have described only some specific by products and their anticancer properties. Only these by products are there in Mediterranean food industry. Please explain.

2. In section 5 authors have described the mechanism of some specific compounds but in figure they have represented the anticancer mode of action of flavonoids which is hard to understand. Please explain any specific if there.

3. Authors have described the role of specific compounds in section 5, if possible summarize the list of these compounds in accordance with their by product source in a table.

4. Authors should add the limitations of the current  approach.

5. Typographical errors are present at several places throughout the manuscript. Authors must proofread the whole manuscript for possible errors.

6. Authors must use full form for abbreviation wherever used first time.

Author Response

Responses to the Reviewer #2 comments

This review article provides an overview about the Mediterranean food industry by-products as a novel source of phytochemicals with a promising role in cancer prevention. The concept is very novel and demanding in context to finding an alternative solution for natural compounds in disease treatment specifically for cancer. Authors described the concept in very well manner, but still manuscript requires some corrections. My suggestions are:

Reply: We would like to thank you very much for your helpful remarks and suggestions. We tried to consider all of them and introduce suitable changes in the revised manuscript. We hope that now the revised version of the manuscript will be recommended for publication. The detailed answers are below.

  1. Authors have described only some specific by products and their anticancer properties. Only these by products are there in Mediterranean food industry. Please explain.

Reply: We agree with the reviewer and we have added a better insight and explanation according to suggestions. Hopefully, this addition has given a slightly better explanation why specific by-products were selected.

  1. In section 5 authors have described the mechanism of some specific compounds but in figure they have represented the anticancer mode of action of flavonoids which is hard to understand. Please explain any specific if there.

Reply: Thank you for this comment, we have added additional text explanations before the figure, as to improve the understanding of the figure itself; “In the Figure 5 a schematic review of flavonoids and some of their main action pathways is given as an example of their mechanism of action in vivo. Through a series of molecular events, flavonoids enter the cells and, by activating some molecules involved in the cell proliferation and cell death, such as for example the tumour suppressor p53, cause further cascade events which ultimately lead to apoptosis of a cell. This underlies the anti-proliferative effect. Another well-known pathway involved in malignant progression is Akt/FAK/Ras/PI3K pathway, that underlies the anti-metastatic effect of flavonoids as well. Another significant pathway in tumor cells is the mitogen-activated protein kinase (MAPK) pathway, which may act anti-inflammatory. These are just some crucial pathways targeted by flavonoids and other signalling pathways in the cell may be triggered by flavonoids as well.”

  1. Authors have described the role of specific compounds in section 5, if possible summarize the list of these compounds in accordance with their by product source in a table.

Reply: Thank you for your suggestion, we believe that the Table 1 already summarizes the main points from the text. The table was positioned at the end of the manuscript and maybe hasn’t been taken into account. Therefore, we have adjusted the position of the Table, so that it can be aligned with the text.

  1. Authors should add the limitations of the current

Reply:  We have added limitations of the current approach at the end of the section Future prospective.

  1. Typographical errors are present at several places throughout the manuscript. Authors must proofread the whole manuscript for possible errors.

Reply: We agree with the reviewer and we have corrected these according to suggestions. 

  1. Authors must use full form for abbreviation wherever used first time.

Reply: Thank you for this comment, we have added now the full form for all abbreviationa wherever used for the first time in the text.

Reviewer 3 Report

The paper gives a very interesting review on the possible use of Food industry by-products with many important references. Only a few changes and clarifications should be done prior to publication:

l.55: unfortunately "the mediterranean diet" can not be clearly defined, but to the knowlegde of the reviewer "small quantities of processed food" is at least misleading: drying, grinding, mincing, fermenting, oil pressing or extracting or even cooking are also a kind of processing frequently used in the  mediterranean diet.

ll.103: toxic organic solvents for extraction can be used in a safely manner (both for human health as well as for the Environment), when the Processes are well designed and controlled.

ll. 119: define "green solvents"

ll. 125: what is meant by eco-friendly: What about the different foot prints, CO2, water,...

ll.140-144: duplication within the text

General: the references and their style has to be checked: sometimes prenames are mentioned (e.g.l.270), sometimes initials, and sometimes only names. 

Author Response

Responses to the comments of Reviewer #3:

The paper gives a very interesting review on the possible use of Food industry by-products with many important references. Only a few changes and clarifications should be done prior to publication:

Reply: We would like to thank you very much for your helpful remarks and suggestions. We tried to consider all of them and introduce suitable changes in the revised manuscript. We hope that now the new version of the manuscript will be recommended for publication. The detailed answers are below.

l.55: unfortunately "the Mediterranean diet" can not be clearly defined, but to the knowledge of the reviewer "small quantities of processed food" is at least misleading: drying, grinding, mincing, fermenting, oil pressing or extracting or even cooking are also a kind of processing frequently used in the  mediterranean diet.

Reply: We agree with the reviewer, and we have specified this point in the text and replaced “small” with “moderate”. Although some processing is performed in the Mediterranean diet, this diet is characterized by lower consumption of processed food in comparison to current dietary habits. Many meals are mainly made of raw foods in combination with basic processed foods, such as olive oil. When it comes to vegetables and fruits, main way of preparation is cooking with no usage of artificial food additives.

,ll.103: toxic organic solvents for extraction can be used in a safely manner (both for human health as well as for the Environment), when the Processes are well designed and controlled.

Reply: Thank you. We have taken your comment into consideration and wrote an additional paragraph (page 3) on the approaches for extractions pointing specifically to toxic organic solvents for extraction and non-conventional more eco-friendly methods.

  1. 119: define "green solvents"

Reply: Thank you for bringing out attention to the term “green solvents”. We have added a new paragraph and additional explanations on page 3 to better define the term itself.

  1. 125: what is meant by eco-friendly: What about the different foot prints, CO2, water,...

Reply:  We have now partially edited the text in which we use the term “eco-friendly”, where we provided a broader explanation of the term itself and the topic of non-conventional methods of extraction with the help of navigation through your suggestions.

ll.140-144: duplication within the text

Reply: Indeed, the duplication is removed now.

General: the references and their style has to be checked: sometimes prenames are mentioned (e.g.l.270), sometimes initials, and sometimes only names.

Reply: We have checked the references and removed the mistakes that have occurred.